# Angiography with optical coherence tomography as a biomarker in multiple sclerosis

Beatriz Cordon[1,2]*, Elisa Vilades[1,2], Elvira Orduna[1,2], María Satue[1,2], Javier Perez-Velilla[1,2], Berta Sebastian[3], Vicente Polo[1,2,4], Jose Manuel Larrosa[1,2,4], Luis Emilio Pablo[1,2,4], Elena Garcia-Martin[1,2,4]

1 Ophthalmology Department, Miguel Servet University Hospital, Zaragoza, Spain, 2 Miguel Servet Ophthalmology Innovation and Research Group (GIMSO), Aragon Institute for Health Research (IIS Aragon), University of Zaragoza, Zaragoza, Spain, 3 Neurology Department, Miguel Servet University Hospital, Zaragoza, Spain, 4 National Ocular Pathology Network (OFTARED) at the Carlos III Institute of Health, Madrid, Spain

* beatrizcordonc@gmail.com

## Abstract

### Purpose

To investigate superficial retinal microvascular plexuses detected by optical coherence tomography angiography (OCT-A) in multiple sclerosis (MS) subjects and compare them with healthy controls.

### Methods

A total of 92 eyes from 92 patients with relapsing-remitting MS and 149 control eyes were included in this prospective observational study. OCT-A imaging was performed using Triton Swept-Source OCT (Topcon Corporation, Japan). The vessel density (VD) percentage in the superficial retinal plexus and optic disc area (6 x 6 mm grid) was measured and compared between groups.

### Results

MS patients showed a significant decrease VD in the superior (p = 0.005), nasal (p = 0.029) and inferior (p = 0.040) parafoveal retina compared with healthy subjects. Patients with disease durations of more than 5 years presented lower VD in the superior (p = 0.002), nasal (p = 0.017) and inferior (p = 0.022) parafoveal areas compared with healthy subjects. Patients with past optic neuritis episodes did not show retinal microvasculature alterations, but patients with an EDSS score of less than 3 showed a significant decrease in nasal (p = 0.024) and superior (p = 0.006) perifoveal VD when compared with healthy subjects.

### Conclusions

MS produces a decrease in retinal vascularization density in the superficial plexus of the parafoveal retina. Alterations in retinal vascularization observed in MS patients are independent of the presence of optic nerve inflammation. OCT-A has the ability to detect subclinical

**Data Availability Statement:** All relevant data are within the manuscript and its Supporting Information files.

**Funding:** The author(s) received no specific funding for this work.

**Competing interests:** The authors have declared that no competing interests exist.

vascular changes and is a potential biomarker for diagnosing the presence and progression of MS.

## Introduction

Multiple sclerosis (MS) is a chronic inflammatory demyelinating autoimmune disease of the central nervous system with axonal degeneration being the main determinant of neurological disability. The etiology is unknown and the course of MS cannot be predicted. Some patients are minimally affected by the disease while in others MS progresses quickly towards total disability. Symptoms of MS will depend on the damaged area and the phenotype. Patients with the relapsing-remitting (RRMS) phenotype suffer symptoms in the form of outbreaks, which can last days or even months, and usually disappear leaving functional sequelae. Some of the most common symptoms are blurred or double vision, muscular weakness, loss of dexterity, numbness or tingling and pain [1,2].

Examination of patients with MS has shown that abnormalities found in the central nervous system are also widespread in the neuro-retina. The visual afferent pathway, from the retina to the visual cortex, is one of the most affected systems, which underlines the importance of studying the visual pathway as a source of potential biomarkers in MS. Inflammation, demyelination and axonal degeneration in the afferent visual pathway are the main cause of visual symptoms in MS.

In recent years, optical coherence tomography (OCT) has demonstrated its ability as a non-invasive way of monitoring neurodegenerative diseases (such as MS, Parkinson's or Alzheimer's) based on quantification of axonal loss in the retina [3–6]. With this technique, the visual pathway is recognized as a model for correlating retinal neurodegeneration and disability in MS and some authors have demonstrated its potential as an early diagnostic tool [7]. In addition, the latest OCT software provides measurements of each retinal layer and even measurements of the choroidal plexus, which seems to provide more accurate measurements of axonal damage and ischemic processes [8–10].

Current diagnostic methods are evolving very rapidly. However, there is still a large time lapse between the first outbreak of the disease and definitive diagnosis. Early diagnosis is related to treatment prescription and therefore a good prognosis for the disease. OCT is a painless, innocuous, and non-invasive imaging test capable of providing high-quality images of the different layers of the retina. In the last 5 years, and due to the high definition of the acquired images, OCT has become able to determine the existence of vascular density (VD) by analyzing contrast in images of blood cells in movement [11]. This finding makes it possible to quickly and easily obtain a 3D cube containing the vascular structures of the retina and the choroid without the need for contrast injection. Based on the results of previous studies on MS and retinal/choroidal vessel density it has been suggested that MS patients show retinal vascular alterations [12–14]. However, research into the use of OCT-A in this field is still scarce [15,16]. This study aims to analyze retinal VD in patients with MS using optical coherence tomography angiography (OCT-A).

## Material and methods

All procedures in this study adhered to the tenets of the Declaration of Helsinki; the experimental protocol was approved by the Ethics Committee of the Miguel Servet Hospital (CEICA), and all participants provided written informed consent to participate in the study.

Based on a preliminary study conducted by our group, we calculated the sample size needed to detect differences of at least 5 μm in the thickness of the CFNR measured by OCT [3], applying a bilateral test with risk α of 5% and risk β of 10% (i.e. with a power of 90%). In order to obtain enough sample of patients with MS, which would allow us to study in depth the natural history of the disease, the non-exposed/exposed ratio was determined to be 0.5. With these data it was concluded that at least 146 eyes would be necessary (73 from healthy subjects and 73 from MS patients). We included more subjects in both groups to improve power of the study.

The study group included only patients with the relapsing-remitting multiple sclerosis phenotype. MS was diagnosed on the basis of the 2010 revision of the McDonald Criteria and was confirmed by a neurologist specializing in MS [17]. The control group consisted of subjects who did not have any type of relevant ocular (epiretinal membranes, glaucoma, age related macular disease etc) or systemic disease previous were related with retinal vascular density such as Diabetes mellitus, arterial hypertension etc. Subjects with visual acuity < 0.4 decimal (6/15 on the Snellen chart), intraocular pressure > 20 mmHg, refractive errors greater than 5 diopters of spherical equivalent refraction or 3 diopters of astigmatism and/or active MS flare (of any neurological deficit) in the 6 months prior to enrollment in the study were excluded from the study. Active MS flare was considered a reason for exclusion because acute axonal loss or neuro-retinal edema could mask neuronal damage secondary to MS progression (i.e., chronic neurodegeneration) or modify retinal VD, which were the main targets of this study. The previous neurological ophthalmological examination was used to detect ocular impairments such as glaucoma, cataract or media opacity that could affect functional vision or retinal microvascularization. During 12 months all subjects were evaluated for best-corrected visual acuity (BCVA), pupillary reflexes, ocular motility, anterior segment examinations, intraocular pressure (IOP) using the Goldmann applanation tonometer, and papillary morphology by funduscopy. Subjects with ocular disease or prior ocular surgery were excluded.

Neurological and ophthalmological examinations were performed less than 3 months before OCT-A data acquisition.

Evaluation of best-corrected visual acuity was performed using the ETDRS optotype, composed of retro-illuminated sheets under photopic light conditions. Patients were seated at a 4-meter distance from the test and best correction was applied until the best possible visual acuity was achieved. The BCVA was expressed in LogMAR. Contrast sensitivity was measured using the Pelli Robson test. This test consists of 6 letters on each line and contrast varies from major to minor. Each line has two different groups, and triplets of letters have the same contrast. It uses a unique spatial frequency of 1 cycle/degree. The test was performed in monocular mode and with the subject at a distance of 1 meter.

Swept-source optical coherence tomography (SS-OCT) (Triton plus; Topcon Corporation, Tokyo, Japan) coupled with non-invasive OCT angiography technology (SS-OCT Angio™) was used to obtain the retinal images. The Topcon SS-OCT uses a tunable laser as a light source to provide a 1050 nm-centered wavelength. This device reaches a scanning speed of 100,000 A-scans per second. Because of this, Topcon SS-OCT visualizes the deepest structures of the retina, detecting even low microvascular density with high sensitivity [18]. All measurements were taken by a single observer, and only images with signal strength index (SSI) and analyzed images with quality score above 50 and 40 respectively [19] were included and also images with movement artefacts were excluded from the analysis.

SS-OCT Angio™ images were acquired using a 6 x 6 mm cube—one cube centered on the fovea and the other centered on the optic disc—with a resolution of 320 x 320. Surface area (SA) was measured using Topcon IMAGEnet® (version 1.19) proprietary software after automated segmentation of the macular area into superficial vascular plexuses (SVP) (SVP-FAZ)

including large vessels. Vessel density values were calculated from the internal limiting membrane to the inner plexiform layer. VD refers to the surface, measured in mm2, which limits the cube in which erythrocyte movement is detected. This software interprets the vascular density of the blood vessels in the scanning area (6 x 6 mm) as the percentage of that area that is occupied by the lumens of the vessels. It provides an ETDRS circular grid (3 mm diameter) which delivers a VD percentage in each of the sections that compose this grid. The grid centered on the fovea divides the macular region into the central foveal area and a perifoveal ring divided into the superior, inferior, nasal and temporal sections. The same grid is transferred to the center of the pit in the optic disc (Fig 1).

Finally, following the Advised Protocol for OCT Study Terminology and Elements (APOSTEL) recommendations for reporting quantitative OCT studies, our study was performed using only one eye in MS patients (randomly selected), except in those patients in which only one eye had a history of optic neuritis (in these cases, both eyes were included in the analysis and were treated as independent) [20].

The neurological evaluation in the MS group included the Expanded Disability Status Scale (EDSS) [21] score, designed by John Kurtzke, which measures patients' functional disability and classifies it into specific ranges that indicate the level of disability presented by the patient: 0–3.5 mild; 4–6 moderate; 7–8 severe; 8.5–9.5 very severe; and 10 death. We also categorized disease duration, prescribed treatments, prior episodes of optic neuritis, and quality of life (QoL) using the multiple sclerosis quality-of-life score (MSQoL-54). This questionnaire is based on a generic survey (the RAND 36-Item Health 1.0) with 18 additional MS-specific items. It consists of a total of 54 items: 52 spread across 12 dimensions (physical health, limitations due to physical problems, limitations due to emotional problems, pain, emotional well-being, energy, perception of health, social function, cognitive function, concern for health, quality of life as a whole, sexual function) plus 2 individual items that measure the change in health status (comparison of current health with that of a year ago) and satisfaction with sexual function. The dimensions are scored from 0 to 100, where a higher value indicates better Health-Related Quality of Life. In addition, two subtotals of mental and physical health are obtained. The MSQoL-54 has been shown to offer good internal consistency, reliability and theoretical validity [22–25].

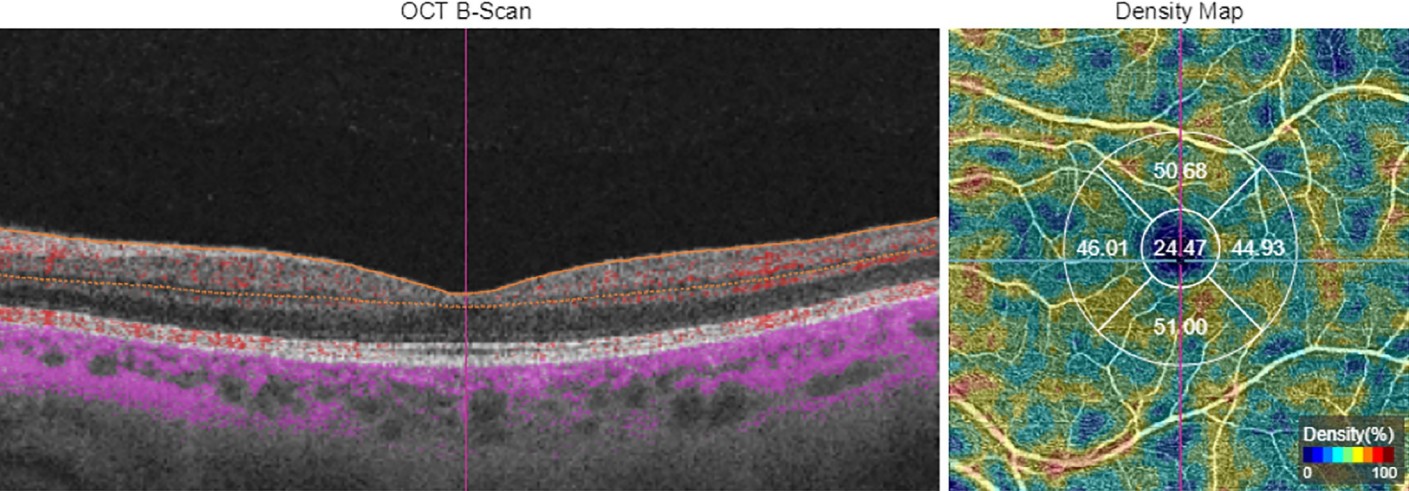

**Fig 1. Image of angiography optical coherence tomography measured in 6x6mm area in superficial vascular plexus with gird centered in macula.** Left image shows a B-scan with orange lines limiting analyzed zone. Right image shows density map with gird divided in five areas; central, nasal, interior, temporal and superior with number of percentages of vessel density.

Statistical analysis: All data studied were recorded in an Excel database and analyzed using the IBM-SPSS statistical package (SPSS Inc, Chicago, IL, USA, version 20.0). The Kolmogorov-Smirnov test showed that most of the study variables were not normally distributed. Thus, non-parametric tests were used in our analysis. Comparison between MS patients and control subjects was performed using the Mann Whitney U test. Differences between groups were analyzed using a one-way ANOVA and a Games-Howell *post hoc* test and significance was set at $p < 0.05$. To avoid a high false positive rate, the Bonferroni correction for multiple tests was calculated and the corrected p values were added to the previously calculated data (see Tables).

## Results

The study group comprised 149 eyes of healthy subjects (20 males and 129 females) and 92 eyes of MS patients (12 males and 80 females). The average age of the sample was 41.76 ± 16.23 years, with no age differences existing between groups (p = 0.955) (Table 1).

MSQoL-54 was completed by 20 MS patients. The scores on the various functional quality-of-life evaluation scales are shown in Table 2.

The structural analysis performed using OCT-A revealed a higher vascular density in the peripapillary area in both groups when compared with the perimacular area. It is necessary to consider that retinal vasculature has some anatomical variations sensitive to axial length, refraction, age... [26], which causes variability in vascular density values. Fig 2 show VD values in control group and EM group related with age (Fig 2).

The MS group showed significant thinning in the ETDRS nasal macular area (p = 0.029), superior macular area (p = 0.005) and inferior macular area (p = 0.040) and a significant reduction in contrast sensitivity (p = 0.008) compared with the control group. Superior macular area remains minimally significantly low vascular density in EM groups despite removing the outlier data (p = 0.047) (Table 3) (Fig 3).

A lower vascular density tendency is observed in the group of MS patients, with significance existing in the nasal, inferior and superior macular areas.

### Analysis by subgroup

Analysis by subgroup was performed on the sample of MS patients based on disease duration, existence or otherwise of previous episodes of optic neuritis and degree of functional impairment by the disease.

**Table 1. Baseline characteristics of the subjects included in the study.**

|  | CONTROL | MS | P |
|---|---|---|---|
| N (eyes) | 149 | 92 | - |
| Eye |  |  | - |
| Right eye (%) | 79 (53) | 46 (50) |  |
| Left eye (%) | 70 (47) | 46 (50) |  |
| Sex |  |  | - |
| Female (%) | 129 (86.6) | 80 (87) |  |
| Male (%) | 20 (13.4) | 12 (13) |  |
| Age (years) | 41.81 ± 18.36 | 41.70 ± 12.11 | - |
| IOP (mmHg) | 15.15 ± 1.57 | 15.37 ± 2.04 | 0.959 |
| EDSS | - | 2.02 ± 1.43 | 0.359 |

Abbreviations: MS, multiple sclerosis; N, number of eyes; IOP, Intraocular pressure; EDSS, Expanded Disability Status Scale.

**Table 2. Mean and standard deviation of MSQoL-54 values in MS patients.**

|  | MS group | |
| --- | --- | --- |
|  | Mean | Mean ± Std. deviation |
| MSQoL-MENT | 69.92 | 26.60 |
| DISTRESS | 64.97 | 32.66 |
| OVERALL | 63.84 | 30.47 |
| EMOTIONAL | 65.27 | 28.87 |
| MENTAL LIMIT | 55.74 | 45.32 |
| COGNITIVE | 57.39 | 34.90 |
| MSQoL-PHY | 65.64 | 21.29 |
| PHYSICAL | 68.63 | 30.49 |
| HEALTH PERCEP | 45.29 | 23.65 |
| ENERGY | 48.08 | 27.86 |
| PHYSICAL LIMIT | 43.70 | 42.67 |
| PAIN | 60.73 | 32.10 |
| SEXUAL FUNC | 56.49 | 35.43 |
| SOCIAL | 66.62 | 35.14 |
| DISTRESS | 64.66 | 33.21 |
| CHANGE | 45.00 | 26.41 |
| SEX SATISF | 70.00 | 29.91 |

To analyze disease duration, one subgroup was created containing patients diagnosed with MS less than or equal to 5 years earlier (44 eyes) and another subgroup was created with patients diagnosed with MS more than 5 years earlier (48 eyes). No significant differences in age, sex or IOP levels were found between the two subgroups. ANOVA analysis with a *post hoc* Games-Howell test found a significant decrease in the nasal macular area (p = 0.001), inferior macular area (p = 0.003) and superior macular area (p < 0.001) in the group that had had the disease for more than 5 years when compared with the healthy subjects. Also, without outliers, superior area macular has a decreased vascular density in EM group. There was also a decrease in contrast sensitivity in both MS groups when compared with healthy subjects (p = 0.026 and p = 0.041) (Table 4).

Two subgroups were also analyzed in the patient group: subgroup MS+ON comprising eyes of patients with a history of optic neuritis, and subgroup MS-ON comprising eyes that had never suffered an episode of neuritis. The *post hoc* analysis indicated a significant decrease in vascular in the nasal and superior macular areas in the MS+ON group in comparison with the control group (p = 0.015, p = 0.033) and in the MS-ON group in comparison with the control group (p = 0.019, p = 0.009). Lower contrast sensitivity was observed in the subgroup with a history of neuritis when compared with the control group (p = 0.004) and with the group of patients with no history of ON (p = 0.036) (Table 5).

Based on the score on the Expanded Disability Status Scale, patients were divided into two groups: one group with EDSS scores of less than 3 and another group with EDSS scores equal to or greater than 3. The ANOVA test and the *post hoc* Games-Howell test for non-parametric tests showed levels of contrast sensitivity and vascular density (nasal macular area, p = 0.003; superior macular area, p = 0.001) that were significantly lower in the group with EDSS scores of less than 3 than among the healthy subjects. However, no differences were observed between the two EDSS groups (Table 6).

The correlation between the EDSS score and quality-of-life questionnaire values was analyzed using Spearman's correlation test, revealing a significant negative correlation between

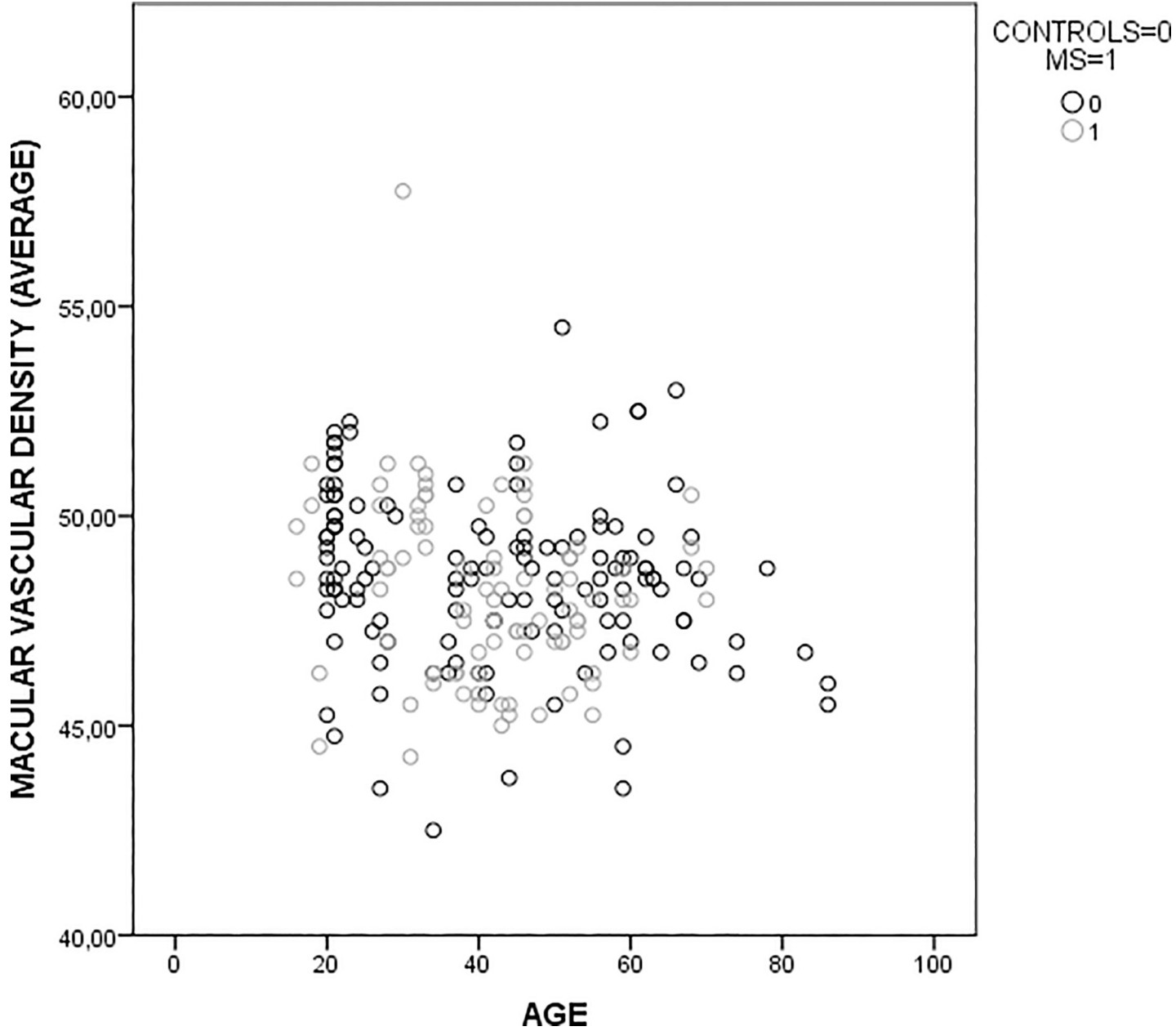

**Fig 2. Scatterplot graphic of vascular density and age in control group and MS group.**

the mental quality-of-life (-0.478), physical limitations (-0.467) and perceived change in comparison with the year before (-0.455). A strongly significant negative correlation was also observed between the EDSS score and physical quality-of-life (Spearman's Correlation p = -0.623) (Table 7).

In addition, the correlation between the EDSS score and VD was analyzed, revealing a negative correlation (-0.271) in VD in the nasal area of the optic disc. (Table 8) and a positive correlation between the inferior VD area of the optic disc and the questions related to mental state in general (Spearman's correlation p = 0.503).

**Table 3. Mean and standard deviation of vascular density.** Average of the vascular density of the four macular quadrants. Best-corrected visual acuity and contrast sensitivity in each group, and significance.

| | | CONTROLS | MS | P |
|---|---|---|---|---|
| | | Mean ± Std. deviation / (without outliers) | Mean ± Std. deviation / (without outliers) | P / (without outliers) |
| BCVA | | 0.05 ± 0.07 | 0.04 ± 0.08 | 0.447 |
| Contrast sensitivity | | 1.86 ± 0.13 | 1.75 ± 0.15 | 0.008* |
| Vascular density (%) | Central macula | 21.89 ± 4.80 / 21.78 ± 3.99 | 21.45 ± 4.51 / 21.47 ± 4.51 | 0.322 / 0.394 |
| | Nasal macula | 48.21 ± 5.52 / 47.20 ± 3.13 | 46.50 ± 2.56 / 46.51 ± 2.57 | 0.029* / 0.104 |
| | Inferior macula | 51.19 ± 5.8 / 49.84 ± 2.99 | 49.50 ± 3.50 / 49.53 ± 3.52 | 0.040* / 0.262 |
| | Temporal macula | 48.24 ± 4.21 / 47.73 ± 3.21 | 47.26 ± 2.46 / 47.24 ± 2.45 | 0.158 / 0.292 |
| | Superior macula | 51.53 ± 5.43 / 50.32 ± 3.17 | 49.51 ± 3.04 / 49.53 ± 3.05 | 0.005* / 0.047 * |
| | Average | 49.79 ± 4.72 | 48.21 ± 2.09 | 0.011* |
| | Central disc | 23.29 ± 12.80 / 23.32 ± 12.73 | 22.59 ± 11.61 / 22.58 ± 11.65 | 0.606 / 0.575 |
| | Nasal disc | 57.98 ± 5.66 / 58.13 ± 5.46 | 57.28 ± 5.34 / 57.29 ± 5.35 | 0.173 / 0.136 |
| | Inferior disc | 65.86 ± 6.27 / 65.97 ± 6.09 | 65.62 ± 7.83 / 65.65 ± 7.84 | 0.870 / 0.851 |
| | Temporal disc | 55.18 ± 6.59 / 55.49 ± 5.15 | 55.57 ± 3.82 / 55.60 ± 3.79 | 0.964 / 0.992 |
| | Superior disc | 64.23 ± 5.73 / 64.47 ± 5.41 | 64.21 ± 4.3 / 64.23 ± 4.32 | 0.379 / 0.342 |

Abbreviations: MS, multiple sclerosis; Std., standard; BCVA, best-corrected visual acuity. Asterisk marks significance based on Mann-Whitney test.

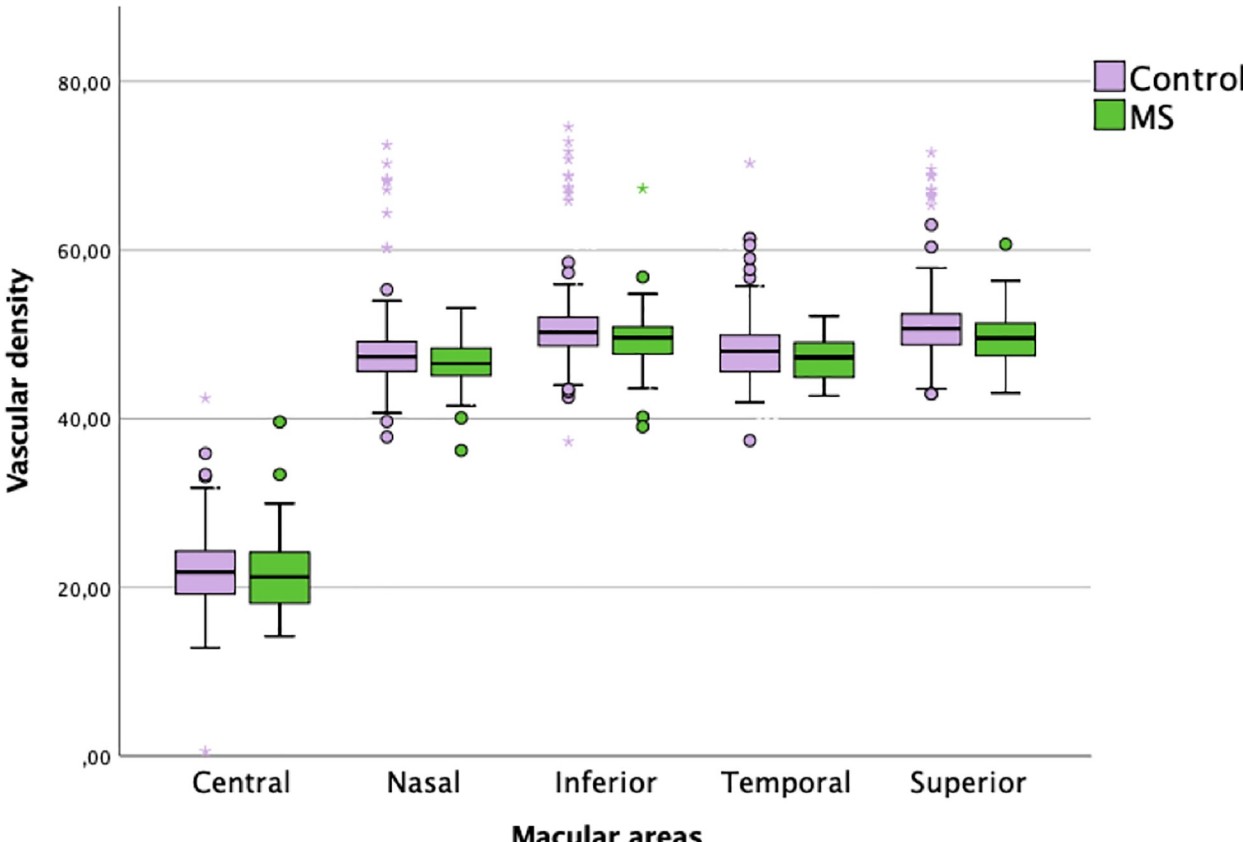

**Fig 3. Graphic of percentage of superficial plexus macular in both groups.**

**Table 4. Mean and standard deviation of vascular density.** BCVA and contrast sensitivity in each group and significance in patients diagnosed with the disease less than 5 years earlier and in patients diagnosed with the disease more than 5 years earlier. Average of the vascular density of the four macular quadrants.

| | | CONTROL | MS ≤ 5 years | MS > 5years | P |
|---|---|---|---|---|---|
| | | Mean ± Std. deviation / (without outliers) | Mean ± Std. deviation / (without outliers) | Mean ± Std. deviation / (without outliers) | P / (without outliers) |
| N | | 145 | 44 | 48 | |
| BCVA | | 0.05 ± 0.07 | 0.02 ± 0.05 | 0.06 ± 0.11 | 0.075 |
| Contrast sensitivity | | 1.86 ± 0.13 | 1.75 ± 0.15 | 1.75 ± 0.15 | 0.029* |
| Vascular density (%) | Central macula | 21.89 ± 4.80 / 21.78 ± 3.99 | 22.35 ± 4.56 / 22.36 ± 4.58 | 20.62 ± 4.34 / 20.65 ± 4.32 | 0.165 / 0.125 |
| | Nasal macula | 48.21 ± 5.52 / 47.20 ± 3.13 | 46.91 ± 2.59 / 46.95 ± 2.60 | 46.14 ± 2.50 / 46.10 ± 2.50 | 0.017* / 0.084 |
| | Inferior macula | 51.19 ± 5.8 / 49.84 ± 2.99 | 50.15 ± 3.65 / 50.16 ± 3.70 | 48.90 ± 3.27 / 48.96 ± 3.27 | 0.022* / 0.155 |
| | Temporal macula | 48.24 ± 4.21 / 47.73 ± 3.21 | 47.72 ± 2.44 / 47.68 ± 2.43 | 46.83 ± 2.42 / 46.83 ± 2.42 | 0.065 / 0.177 |
| | Superior macula | 51.53 ± 5.43 / 50.32 ± 3.17 | 50.18 ± 2.69 / 50.20 ± 2.69 | 48.89 ± 3.24 / 48.92 ± 3.25 | 0.002* / 0.025* |
| | Average | 49.79 ± 4.72 | 48.74 ± 3.21 | 47.69 ± 3.52 | 0.014* |
| | Central disc | 23.29 ± 12.80 / 23.32 ± 12.73 | 23.47 ± 12.16 / 23.50 ± 12.21 | 21.68 ± 11.06 / 21.62 ± 11.11 | 0.735 / 0.708 |
| | Nasal disc | 57.98 ± 5.66 / 58.13 ± 5.46 | 57.57 ± 4.86 / 57.61 ± 4.86 | 56.97 ± 5.85 / 56.95 ± 5.86 | 0.577 / 0.448 |
| | Inferior disc | 65.86 ± 6.27 / 65.97 ± 6.09 | 66.96 ± 5.27 / 66.98 ± 5.25 | 64.22 ± 9.70 / 64.26 ± 9.72 | 0.175 / 0.168 |
| | Temporal disc | 55.18 ± 6.59 / 55.49 ± 5.15 | 56.49 ± 3.68 / 56.57 ± 3.64 | 54.61 ± 3.78 / 54.60 ± 3.73 | 0.277 / 0.146 |
| | Superior disc | 64.23 ± 5.73 / 64.47 ± 5.41 | 64.69 ± 4.31 / 64.70 ± 4.40 | 63.71 ± 4.28 / 63.74 ± 4.24 | 0.687 / 0.636 |

Abbreviations: MS, multiple sclerosis; Std., standard; BCVA, best-corrected visual acuity. Asterisk marks significance based on ANOVA and posterior post hoc Games-Howell test.

**Table 5. Mean and standard deviation of vascular density.** Average of the vascular density of the four macular quadrants. Best-corrected visual acuity and contrast sensitivity in MS group with past optic neuritis (MS+ON) and MS group without optic neuritis (MS-ON).

| | | CONTROL | MS-ON | MS+ON | P |
|---|---|---|---|---|---|
| | | Mean ± Std. deviation / (without outliers) | Mean ± Std. deviation / (without outliers) | Mean ± Std. deviation / (without outliers) | P / (without outliers) |
| N | | 151 | 72 | 20 | |
| BCVA | | 0.05 ± 0.07 | 002 ± 0.06 | 0.07 ± 0.12 | 0.177 |
| Contrast sensitivity | | 1.83 ± 0.14 | 1.78 ± 0.15 | 1.68 ± 0.11 | 0.016* |
| Vascular density (%) | Central macula | 21.89 ± 4.80 / 21.78 ± 3.99 | 21.60 ± 4.81 / 21.62 ± 4.82 | 20.80 ± 3.62 / 20.80 ± 3.59 | 0.683 / 0.697 |
| | Nasal macula | 48.21 ± 5.52 / 47.20 ± 3.13 | 46.65 ± 2.45 / 46.66 ± 2.46 | 46.15 ± 1.95 / 46.13 ± 1.92 | 0.046* / 0.305 |
| | Inferior macula | 51.19 ± 5.8 / 49.84 ± 2.99 | 49.74 ± 3.74 / 49.79 ± 3.76 | 48.56 ± 2.77 / 48.53 ± 2.75 | 0.053 / 0.336 |
| | Temporal macula | 48.24 ± 4.21 / 47.73 ± 3.21 | 47.32 ± 2.57 / 47.31 ± 2.56 | 47.00 ± 2.20 / 46.93 ± 2.22 | 0.147 / 0.464 |
| | Superior macula | 51.53 ± 5.43 / 50.32 ± 3.17 | 49.73 ± 3.01 / 49.75 ± 3.04 | 48.78 ± 3.37 / 48.87 ± 3.27 | 0.012* / 0.193 |
| | Average | 49.79 ± 4.72 | 48.39 ± 2.15 | 47.61 ±2.06 | 0.021* |
| | Central disc | 23.29 ± 12.80 / 23.32 ± 12.73 | 21.05 ± 10.87 / 21.03 ± 10.95 | 22.32 ± 10.46 / 22.33 ± 10.35 | 0.281 / 0.265 |
| | Nasal disc | 57.98 ± 5.66 / 58.13 ± 5.46 | 58.10 ± 5.06 / 58.12 ± 5.06 | 55.60 ± 5.67 / 55.53 ± 5.63 | 0.288 / 0.233 |
| | Inferior disc | 65.86 ± 6.27 / 65.97 ± 6.09 | 65.52 ± 8.65 / 65.55 ± 8.66 | 65.13 ± 4.38 / 65.13 ± 4.32 | 0.854 / 0.809 |
| | Temporal disc | 55.18 ± 6.59 / 55.49 ± 5.15 | 56.04 ± 3.71 / 56.08 ± 3.67 | 53.81 ± 4.19 / 53.80 ± 4.13 | 0.336 / 0.228 |
| | Superior disc | 64.23 ± 5.73 / 64.47 ± 5.41 | 64.50 ± 4.28 / 64.54 ± 4.30 | 64.07 ± 3.82 / 64.07 ± 3.83 | 0.882 / 0.936 |

Abbreviations: MS, multiple sclerosis; Std., standard; BCVA, best-corrected visual acuity; ON, optic neuritis. Asterisk marks significance based on ANOVA and posterior post hoc Games-Howell test.

**Table 6. Mean and standard deviation of vascular density.** Average of the vascular density of the four macular quadrants. Best-corrected visual acuity and contrast sensitivity in control subjects. MS group with EDSS of less than 3 and MS group with EDSS of more than 3.

| | | CONTROLS | MS EDSS < 3 | MS EDSS ≥ 3 | P |
|---|---|---|---|---|---|
| | | Mean ± Std. deviation / (without outliers) | Mean ± Std. deviation / (without outliers) | Mean ± Std. deviation / (without outliers) | P / (without outliers) |
| N | | 151 | 56 | 30 | – |
| BCVA | | 0.04 ± 0.07 | 0.04 ± 0.08 | 0.05 ± 0.1 | 0.930 |
| Contrast sensitivity | | 1.84 ± 0.13 | 1.74 ± 0.16 | 1.76 ± 0.14 | 0.027* |
| Vascular density (%) | Central macula | 21.89 ± 4.80 / 21.78 ± 3.99 | 21.36 ± 4.82 / 21.39 ± 4.84 | 21.66 ± 4.08 / 21.67 ± 4.03 | 0.791 / 0.859 |
| | Nasal macula | 48.21 ± 5.52 / 47.20 ± 3.13 | 46.22 ± 2.76 / 46.23 ± 2.80 | 47.03 ± 2.14 / 47.03 ± 2.11 | 0.024* / 0.126 |
| | Inferior macula | 51.19 ± 5.8 / 49.84 ± 2.99 | 49.59 ± 3.69 / 49.63 ± 3.72 | 49.46 ± 3.46 / 49.53 ± 3.45 | 0.078 / 0.894 |
| | Temporal macula | 48.24 ± 4.21 / 47.73 ± 3.21 | 47.21 ± 2.31 / 47.21 ± 2.33 | 47.43 ± 2.72 / 47.40 ± 2.70 | 0.185 / 0.575 |
| | Superior macula | 51.53 ± 5.43 / 50.32 ± 3.17 | 49.29 ± 3.08 / 49.39 ± 3.07 | 49.76 ± 3.16 / 49.67 ± 3.21 | 0.006* / 0.148 |
| | Average | 49.79 ± 4.72 | 48.08 ± 3.44 | 48.42 ± 3.09 | 0.057 |
| | Central disc | 23.29 ± 12.80 / 23.32 ± 12.73 | 22.19 ± 11.96 / 22.20 ± 11.95 | 24.47 ± 11.89 / 24.42 ± 12.07 | 0.744 / 0.752 |
| | Nasal disc | 57.98 ± 5.66 / 58.13 ± 5.46 | 57.67 ± 5.25 / 57.68 ± 5.21 | 55.19 ± 4.94 / 55.25 ± 5.09 | 0.053 / 0.038* |
| | Inferior disc | 65.86 ± 6.27 / 65.97 ± 6.09 | 66.48 ± 5.18 / 66.48 ± 5.18 | 62.72 ± 12.15 / 62.79 ± 12.16 | 0.064 / 0.061 |
| | Temporal disc | 55.18 ± 6.59 / 55.49 ± 5.15 | 55.08 ± 4.11 / 55.11 ± 4.06 | 56.28 ± 3.30 / 56.33 ± 3.31 | 0.674 / 0.559 |
| | Superior disc | 64.23 ± 5.73 / 64.47 ± 5.41 | 64.62 ± 4.37 / 64.64 ± 4.41 | 63.36 ± 4.45 / 63.37 ± 4.31 | 0.617 / 0.569 |

Abbreviations: MS, multiple sclerosis; Std., standard. Asterisk marks significance based on ANOVA and posterior post hoc Games-Howell test.

**Table 7. Spearman's correlation between MSQoL-54 values and EDSS.**

| Spearman's correlation | EDSS | Macular | | | | | Disc | | | | |
|---|---|---|---|---|---|---|---|---|---|---|---|
| | | Central | Nasal | Inferior | Temporal | Superior | Central | Nasal | Inferior | Temporal | Superior |
| **MSQoL-MENT** | -0.478* | 0.450* | -0.204 | -0.235 | -0.044 | -0.090 | -0.162 | -0.326 | -0.110 | -0.068 | -0.095 |
| DISTRESS | -0.216 | 0.015 | -0.066 | -0.132 | -0.239 | -0.224 | -0.079 | 0.273 | 0.257 | -0.026 | 0.223 |
| OVERALL | -0.129 | 0.023 | 0.115 | -0.002 | -0.038 | -0.254 | -0.119 | 0.203 | 0.503* | 0.086 | 0.343 |
| EMOTIONAL | -0.138 | -0.06 | 0.053 | -0.038 | -0.181 | -0.294 | 0.036 | 0.083 | 0.202 | -0.013 | 0.080 |
| MENTAL LIMIT | -0.141 | 0.367 | 0.063 | 0.106 | -0.274 | -0.086 | 0.040 | -0.019 | 0.234 | 0.307 | -0.022 |
| COGNITIVE | -0.227 | -0.289 | 0.122 | 0.010 | 0.049 | -0.219 | -0.062 | 0.277 | 0.216 | -0.040 | 0.109 |
| **MSQoL-PHI** | -0.623** | 0.352 | -0.296 | -0.337 | -0.038 | -0.208 | -0.081 | -0.313 | -0.126 | -0.223 | 0.114 |
| PHYSICAL | -0.321 | -0.011 | -0.374 | -0.229 | -0.335 | -0.336 | 0.122 | 0.051 | 0.062 | -0.154 | 0.223 |
| HEALTH PERCEP | -0.157 | -0.199 | -0.172 | -0.219 | -0.174 | -0.267 | -0.063 | -0.006 | 0.123 | -0.100 | 0.180 |
| ENERGY | -0.253 | -0.053 | -0.016 | -0.168 | -0.357 | -0.386 | 0.097 | 0.131 | 0.101 | -0.100 | 0.0132 |
| PHYSICAL LIMIT | -0.467* | 0.087 | 0.009 | -0.126 | -0.094 | -0.359 | 0.307 | 0.213 | 0.240 | -0.106 | 0.269 |
| PAIN | -0.040 | -0.118 | -0.110 | -0.002 | -0.132 | -0.143 | 0.097 | -0.057 | 0.144 | 0.134 | 0.087 |
| SEXUAL FUNC | -0.159 | 0.107 | 0.107 | 0.156 | -0.053 | -0.159 | 0.030 | 0.311 | 0.402 | 0.241 | 0.125 |
| SOCIAL | -0.161 | -0.033 | 0.063 | -0.026 | -0.173 | -0.286 | -0.017 | 0.044 | 0.292 | 0.099 | 0.291 |
| DISTRESS | -0.216 | 0.015 | -0.066 | -0.132 | -0.239 | -0.224 | -0.079 | 0.273 | 0.257 | -0.026 | 0.223 |
| **CHANGE** | -0.455* | 0.351 | 0.031 | -0.116 | -0.0147 | -0.204 | -0.203 | 0.115 | 0.184 | -0.084 | -0.101 |
| **SEX SATISF** | -0.157 | 0.314 | -0.132 | 0.008 | 0.150 | 0.148 | -0.091 | 0.065 | 0.364 | 0.228 | 0.178 |

Asterisk marks significance based on Bonferroni correction for multiple comparisons.

**Table 8. Spearman correlation between vascular density, OCTA values and EDSS.**

|  | EDSS Spearman's Correlation |
|---|---|
| Macular (C) | -0.006 |
| Macular (N) | 0.194 |
| Macular (I) | 0.033 |
| Macular (T) | 0.032 |
| Macular (S) | 0.072 |
| Disc (C) | 0.184 |
| Disc (N) | -0.271* |
| Disc (I) | -0.211 |
| Disc (T) | 0.074 |
| Disc (S) | -0.093 |

Asterisk marks significance based on Bonferroni correction for multiple comparisons.

## Discussion

Several studies have found that brain blood density is significantly affected in both early-diagnosed RRMS and primary progressive MS, indicating that this affectation is already present in the early stages of the pathology [27,28]. Animal studies have shown that chronic hypoperfusion of the brain can induce neurodegenerative changes, including the axonal degeneration so characteristic of this disease [29].

In this study, a significant decrease in VD was observed in the superior, nasal and inferior macular areas in patients suffering from MS with Triton Plus OCTA. More precisely, VD decrease remains significant when we remove outliers. Our results support previous research, such as Lanzillo R et al [12] (with Octovue OCTA device) and Olwen C Murphy et al [30] (Spectralis OCTA), both showing a loss of VD in SVP macular ETDRS in MS patients. Only one study evidenced an increase in VD in MS patients [31], which was realized with Angioplex Zeiss device.

Other studies as García-Martín et al [32], conduct five-year follow-up of MS patients, observed that the retinal nerve fiber layer and the ganglion cell layer measured in the peripapillary ring are thinner in MS patients, especially in the temporal and superior areas. In light of these results, it may be hypothesized that a relationship exists between the decrease in the temporal and superior peripapillary RNFL and vasculature that nourishes this layer, especially in superior macular area that shows a marked decrease in our results. Feucht et al [33] in his study shows this with a high correlation between loss retinal vasculature and thinning retinal layers. However, there are still very few published studies on OCT-A and MS, and it is still not known whether retinal degeneration occurs due to the decrease in VD or whether it is an unrelated phenomenon that could precede the disease or occur simultaneously with it [34]. Follow-up studies of MS patients could determine the relationship between neuronal loss and VD in the retina. Other neurodegenerative diseases such as Alzheimer's also produced similar decreases in VD in the superficial plexus of the retina when measured using OCT-A, which suggests that this test is able to detect ophthalmological impairments that occur in the course of neurodegenerative diseases [35].

Our study shows that several macular areas present VD less in patients who have had the disease for more than 5 years when compared with healthy subjects, indicating that the advance of MS is reflected at the level of the superficial plexus of the retina in the form of macular level impairment. There was a strong negative correlation between the quality-of-life values and the EDSS score. At the same time, structural analysis by subgroup based on the EDSS reveals that the group with scores of less than 3 exhibited less VD in the superior and nasal

areas than healthy subjects. These results suggest that OCT-A is able to detect a reduction in retinal VD in patients with a mild disability and that perhaps this vascular impairment is not as marked in subjects with more severe cases of the disease. It would be revealing to evaluate this test in subjects with clinically isolated syndrome and to observe whether OCT-A is able to anticipate diagnosis of MS. There was, however, no significant correlation between either VD and EDSS values or the quality-of-life questionnaire scores.

The results of this study show that retinal microvascularization is not impaired in patients who have suffered episodes of optic neuritis when compared with patients who have not suffered neuritis. However, contrast sensitivity is diminished after an episode of neuritis, as is reported in the literature [36]. The sample of eyes with prior ON is small—just 20 affected eyes—meaning that a study with a greater number of patients who have suffered this disease is required. Structural disorders in these patients, highly evidenced in several studies, are related to the inner layers of the retina, which leads to the hypothesis that VD in the intermediate layers is not affected by the inflammatory episode. Likewise, the macular area is not particularly damaged by ON. Therefore, our results suggest, in line with other previous studies [37], that in the course of MS the vascularization of the retina is not exclusively affected by inflammatory lesion of the optic nerve, but rather by the disease itself.

Our study has some limitations. In OCT-A, proper segmentation and analysis of the vascular plexuses require high-quality images. This presents a challenge in patients with debilitating diseases, especially in those suffering severe cognitive and physical impairments that make it difficult for them to follow instructions and maintain concentration and this affected the small sample size presented in this study. Additionally, the number of patients who answered MSQoL-54 was small compared to the total sample. Another limitation is the fact that the ETDRS grid needs to be positioned manually by the operator using the zone of minimum central value for the VD as the point of reference. The study did not consider analyze immunotherapy treatment that patient taken which could alter the results. The optic disc presents a wide range of anatomical variations within the parameters of normality. Therefore, although patients with atypical eye characteristics were excluded, it should be noted that vascular data may be conditioned by the characteristics of each optic nerve. We expose a retinal vessel density loss in FM patients without consider structural retinal state, this device does not measure thickness in same mensuration of OCTA, but more information about thickness of the retinal layers would complete the study.

In conclusion, our study shows that there is microvascular impairment in the perifoveal retina of MS patients. This decrease in VD is detectable in patients with mild functional impairment and is most evident in those who have had the disease for longer. OCT-A is a rapid, simple, non-invasive and easy-to-manage test that is useful in diagnosing the disease as it flags decreased VD values even in patients with mild disabilities. Although the analysis area is still small and we can only quantify the superficial plexus, the rapid development of optical coherence tomography will eventually remove these limitations.

## Supporting information

**S1 Data.**
(XLSX)

## Author Contributions

**Conceptualization:** Beatriz Cordon, Elisa Vilades, Elvira Orduna, María Satue, Berta Sebastian, Vicente Polo, Luis Emilio Pablo, Elena Garcia-Martin.

**Data curation:** Beatriz Cordon, Elisa Vilades, Elvira Orduna, Elena Garcia-Martin.

**Formal analysis:** Beatriz Cordon, Elisa Vilades, Elvira Orduna, María Satue, Elena Garcia-Martin.

**Funding acquisition:** Beatriz Cordon.

**Investigation:** Beatriz Cordon, Elisa Vilades, Elvira Orduna, María Satue, Javier Perez-Velilla, Berta Sebastian, Vicente Polo, Jose Manuel Larrosa, Luis Emilio Pablo, Elena Garcia-Martin.

**Methodology:** Beatriz Cordon, Elisa Vilades, Elvira Orduna, Elena Garcia-Martin.

**Project administration:** Beatriz Cordon, Elisa Vilades, María Satue.

**Resources:** Beatriz Cordon, Elisa Vilades, Elvira Orduna, María Satue.

**Software:** Beatriz Cordon, Elisa Vilades, Elvira Orduna, María Satue, Elena Garcia-Martin.

**Supervision:** Beatriz Cordon, Elisa Vilades, Elvira Orduna, María Satue, Javier Perez-Velilla, Berta Sebastian, Vicente Polo, Jose Manuel Larrosa, Luis Emilio Pablo, Elena Garcia-Martin.

**Validation:** Beatriz Cordon, Elisa Vilades, María Satue, Javier Perez-Velilla, Berta Sebastian, Vicente Polo, Jose Manuel Larrosa, Luis Emilio Pablo, Elena Garcia-Martin.

**Visualization:** Beatriz Cordon, Elisa Vilades, María Satue, Javier Perez-Velilla, Berta Sebastian, Vicente Polo, Jose Manuel Larrosa, Luis Emilio Pablo, Elena Garcia-Martin.

**Writing – original draft:** Beatriz Cordon, María Satue, Elena Garcia-Martin.

**Writing – review & editing:** Beatriz Cordon, Elisa Vilades, Elvira Orduna, María Satue, Javier Perez-Velilla, Berta Sebastian, Vicente Polo, Jose Manuel Larrosa, Luis Emilio Pablo, Elena Garcia-Martin.

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
