## [Decision Letter · Decision Letter 0]

26 May 2020

PONE-D-20-14099

Angiography with optical coherence tomography as a biomarker in multiple sclerosis

PLOS ONE

Dear Dr. Ciordia,

Thank you for submitting your manuscript to PLOS ONE. After careful consideration, we feel that it has merit but does not fully meet PLOS ONE’s publication criteria as it currently stands. Therefore, we invite you to submit a revised version of the manuscript that addresses the points raised during the review process.

We look forward to receiving your revised manuscript.

Kind regards,

Ireneusz Grulkowski, PhD

Academic Editor

PLOS ONE

Journal Requirements:

2. Please ensure that you refer to Figure 1 in your text as, if accepted, production will need this reference to link the reader to the figure.

Reviewers' comments:

Reviewer's Responses to Questions

**Comments to the Author**

1. Is the manuscript technically sound, and do the data support the conclusions?

Reviewer #1: Yes

Reviewer #2: Partly

Reviewer #3: Yes

2. Has the statistical analysis been performed appropriately and rigorously? 

Reviewer #1: Yes

Reviewer #2: No

Reviewer #3: Yes

3. Have the authors made all data underlying the findings in their manuscript fully available?

Reviewer #1: Yes

Reviewer #2: Yes

Reviewer #3: Yes

4. Is the manuscript presented in an intelligible fashion and written in standard English?

Reviewer #1: Yes

Reviewer #2: No

Reviewer #3: Yes

5. Review Comments to the Author

Reviewer #1: the authors had conducted a well designed informative study. I congratulate authors for this work.

however there are some issues should be reviewed.

1- Abstract- no comment

2- Introduction - the sentence in line 90 which begins with "Based on..." needs references. Suggestions: pubmedID: 28933233, 31857713,28814415

3- materials and methods- line 134- describe the low quality images, which parameters the author used to determine the image quality? Did the author set a degree for SSI quality to include the scans in the study?

- Does the SS-OCTA device software has the ability to remove the large vessels automatically from the peripapillary images? If it does not, the authors should add this information because it is very well known that the diseases such as glaucoma, MS, Alzheimer ...etc do not effect the large vessels untill the late stages of these diseases. the authors also should discuss this issue at the discussion.

-the time period of the recruitment of the study subjects is missing.

-the power/sample size calculations are missing which are very important for prospective studies. If the sample size had not been calculated by the authors prior to the study, the authors should declare this information like "pawer calculations were not executed as the study was exploratory.

4- results - no comment

5- discussion - the authors had not included the immunotherapy that the MS patients get in to the study, which could alter the result. This issue should be discussed and given as a limitations of the study.

Reviewer #2: Dr. Cordon et al. studied retinal microvascular alterations of patients with MS using OCTA. They tried to determine if OCTA measurements could be the biomarkers for MS. While the results may add the contribution to the filed, some issues will need to be addressed.

1. OCTA does not measure blood flow (meaning volume per time), but vessel structure. The results should be clearly defined, which is vessel density (the area occupied by the vessels in percentage). Please remove the term of blood flow or vascular area flow, which are wrong expression.

2. Line 88: ref 11 is for retino-neuron structure, not for vascular flow. Please double check

3. Published papers regarding OCTA on MS is more than ref 12,13. The authors need to review and discuss the previously published papers from Feucht, Lanzillo, Wang and Jiang.

4. Line 102: “The control group consisted of subjects who did not have any type of relevant ocular or systemic disease.” The meaning of this sentence is not clear to the reviewer.

5. Line 116-117: “neurological and ophthalmologial examinations were…” should be combined into line 110 to clarify the meaning of “previous”.

6. It is not clear to this reviewer whether the analysis was based on 6 x 6 mm scan for the area of 6 mm circular area with DTDRS partition, or for the area of 3 mm circular area. If this was based on the 3 mm ETDRS partition, why did the scan protocol use 6 x 6 mm. The authors will need to include a figure to show the scan area and partition.

7. Since both eyes in some cases were included, GEE will be needed to analyze the difference of the measurements between groups. If both eyes of the patient were treated independently, the statistic power was inflated, which is not right.

8. The authors need to report the cut off value of scan quality scores.

9. The superficial vascular plexuses (SVP) appeared to be defined as the vasculature from ILM to RPE, which is the vascular network of the entire retina, not the only SVP. The authors will need to separate SVP and deep vascular plexus for analysis and comparison. In addition, the authors will need to include OCTA enface view image and cross-sectional structural image with the segmented boundaries.

10. Since the OCTA can get the thickness information of intraretinal layers, the authors need to report the thickness information to show whether structural thinning occurred as the author discussed this relation in Discussion.

11. Again, if individual thinning of the retina occurred, the loss of vessels may be proportional to the loss of tissue. Therefore, normalization of the vessel loss needs to be done by the vessel density divided by the corresponding tissue volume. This will provide more accurate information whether hypoperfusion occurs. The absolute value decrease of the vessel percentage without knowing the perfused tissue mess in volume does not mean hypoperfusion.

12. The authors will need to discuss the mechanism of localized alterations of the VD in the whole group and subgroups. Previous studies showed more severe loss of VD in MS+ON. It is surprised that there appeared to be no difference of the VD measurements between MS+ON and MS-ON. The authors explained this to be due to small sample size. Increasing the sample with MS+ON will be needed.

Reviewer #3: Beatriz Cordón Ciordia and colleagues provide a paper on OCTA vessel density in patients with MS and compared their data with normals.

Introduction and method section is finely written. The authors should include here exemplary vessel density en face pictures with an overlay of the grit used to better understand the position of the values. Especially central macula should be highlighted as normally this area is the foveal avascular zone without any vessels.

The results are well displayed and correlated. The graph should show the results in boxplots to better visualize the distribution of the results in normals and MS patients.

The discussion should include an explanation why nasal and superior sectors might be correlating so well and disctinct from the normal population. In MS RNFL in the papillomacular bundle shows loss of thickness in structural OCT already in very early stages. These axons belong mainly to midget or P-ganglion cells. Does the OCTA device provide structural measures of the areas measured with OCTA? if yes, the authors should include these measurements and correlate them too.

The authors should correlate vessel density measures with their data as most studies were performed with Optovue and this study was done with a Topcon device. As all OCTA devices use different algorithms and the data are hardly comparable the authors should also take this methodological aspect into account.

6. PLOS authors have the option to publish the peer review history of their article (what does this mean?). If published, this will include your full peer review and any attached files.

Reviewer #1: No

Reviewer #2: No

Reviewer #3: No

---

## [Author Response · Author response to Decision Letter 0]

7 Jul 2020

June 30, 2020

Dear Editor-in-Chief,

Please find enclose the manuscript entitled “Angiography with optical coherence tomography as a biomarker in multiple sclerosis (PONE-D-20-14099)”. We have improved the paper in response to your suggestions, and we would like it to be considered for publication in the Plos One.

As the corresponding author I confirm that all authors have read the revision and agree with changes. 

We thank reviewers for providing us with very pertinent and helpful comments.

RESPONSES TO REVIEWERS

Reviewer #1 

The authors had conducted a well designed informative study. I congratulate authors for this work.however there are some issues should be reviewed.

1- Abstract- no comment.

2- Introduction - the sentence in line 90 which begins with "Based on..." needs references. Suggestions: pubmedID: 28933233, 31857713,28814415.

Response: New references has been added of more studies about MS and OCTA in lines 89-91 and line 300.

-Based on the results of previous studies on MS and retinal/choroidal vessel density it has been suggested that MS patients show retinal vascular alterations [12-14]. 

12-Lanzillo R, Cennamo G, Criscuolo C, Carotenuto A, Velotti N, Sparnelli F, et al. Optical coherence tomography angiography retinal vascular network assessment in multiple sclerosis. Mult Scler. 2018;24(13):1706-1714.

13-Yilmaz H, Ersoy A, Icel E. Assessments of vessel density and foveal avascular zone metrics in multiple sclerosis: an optical coherence tomography angiography study. Eye (Lond). 2020; 34:771-778.

14-Spain RI, Liu L, Zhang X, Jia Y, Tan O, Bourdette D, et al. Optical coherence tomography angiography enhances the detection of optic nerve damage in multiple sclerosis. Br J Ophthalmol. 2018; 102:520-524

-In light of these results, it may be hypothesized that a relationship exists between the decrease in the temporal and superior peripapillary RNFL and vasculature that nourishes this layer like Feucht et al [28] in his study that shows a high correlation between loss retinal vasculature and thinning retinal layers.

28-Feucht, N., Maier, M., Lepennetier, G., Pettenkofer, M., Wetzlmair, C., Daltrozzo, T, et al. Optical coherence tomography angiography indicates associations of the retinal vascular network and disease activity in multiple sclerosis. Multiple Sclerosis Journal. 2019;25(2), 224–234.

3- materials and methods- line 134- describe the low quality images, which parameters the author used to determine the image quality? Did the author set a degree for SSI quality to include the scans in the study?

Response: information added in methodology line 141-144.

-All measurements were taken by a single observer and low-quality ,and only images with signal strength index (SSI) and analyzed images with quality score above 50 and 40 respectively [19] were included and also images with movement artefacts were excluded from the analysis.

19. Tang, FY, Erica O.Chan, Sun Z, Wong R, Lok J, Szeto S et al. Clinically relevant factors associated with quantitative optical coherence tomography angiography metrics in deep capillary plexus in patients with diabetes. Eye and vision. 2020;7:7.

- Does the SS-OCTA device software has the ability to remove the large vessels automatically from the peripapillary images? If it does not, the authors should add this information because it is very well known that the diseases such as glaucoma, MS, Alzheimer ...etc do not effect the large vessels untill the late stages of these diseases. the authors also should discuss this issue at the discussion.

Response: Software did not include the ability to remove large vessels automatically which makes our results more consistent because we have measured microvasculature. Also, we have added this explanation in methodology Line 146-149

-Surface area (SA) was measured using Topcon IMAGEnet® (version 1.19) proprietary software after automated segmentation of the macular area into superficial vascular plexuses (SVP) (SVP-FAZ) including large vessels.

-the time period of the recruitment of the study subjects is missing.

Response: information added in methodology line 121.

-During 12 months all subjects were evaluated for best-corrected visual acuity (BCVA), pupillary reflexes, ocular motility, anterior segment examinations, intraocular pressure (IOP) using the Goldmann applanation tonometer, and papillary morphology by funduscopy.

-the power/sample size calculations are missing which are very important for prospective studies. If the sample size had not been calculated by the authors prior to the study, the authors should declare this information like "power calculations were not executed as the study was exploratory.

Response: We added this information in paragraph 2 of method section (Line 100-106): 

-Based on a preliminary study conducted by our group, we calculated the sample size needed to detect differences of at least 5 μm in the thickness of the CFNR measured by OCT [3], applying a bilateral test with risk α of 5% and risk β of 10% (i.e. with a power of 90%). In order to obtain enough sample of patients with MS, which would allow us to study in depth the natural history of the disease, the non-exposed/exposed ratio was determined to be 0.5. With these data it was concluded that at least 146 eyes would be necessary (73 from healthy subjects and 73 from MS patients). We included more subjects in both groups to improve power of the study.

4- results - no comment

5- discussion - the authors had not included the immunotherapy that the MS patients get in to the study, which could alter the result. This issue should be discussed and given as a limitations of the study.

Response: information added in discussion line 344-345.

-The study did not consider immunotherapy treatment that patient taken which could alter the results.

Reviewer #2: 

Dr. Cordon et al. studied retinal microvascular alterations of patients with MS using OCTA. They tried to determine if OCTA measurements could be the biomarkers for MS. While the results may add the contribution to the filed, some issues will need to be addressed.

1. OCTA does not measure blood flow (meaning volume per time), but vessel structure. The results should be clearly defined, which is vessel density (the area occupied by the vessels in percentage). Please remove the term of blood flow or vascular area flow, which are wrong expression.

Response: We changed term blood flow and added vascular density (VD) to be more exactly with the concept.

2. Line 88: ref 11 is for retino-neuron structure, not for vascular flow. Please double check

Response: Reference 11 was reviewed and referenced correctly

-11-de Carlo TE, Romano A, Waheed NK, Duker JS. A review of optical coherence tomography angiography (OCTA). Int J Retina Vitreous. 2015;1:5. doi:10.1186/s40942-015-0005-8.

3. Published papers regarding OCTA on MS is more than ref 12,13. The authors need to review and discuss the previously published papers from Feucht, Lanzillo, Wang and Jiang.

Response: New information added of more studies about MS and OCTA in lines 89-91 and line 300.

-Based on the results of previous studies on MS and retinal/choroidal vessel density it has been suggested that MS patients show retinal vascular alterations [12-14]. 

12-Lanzillo R, Cennamo G, Criscuolo C, Carotenuto A, Velotti N, Sparnelli F, et al. Optical coherence tomography angiography retinal vascular network assessment in multiple sclerosis. Mult Scler. 2018;24(13):1706-1714.

13-Yilmaz H, Ersoy A, Icel E. Assessments of vessel density and foveal avascular zone metrics in multiple sclerosis: an optical coherence tomography angiography study. Eye (Lond). 2020; 34:771-778.

14-Spain RI, Liu L, Zhang X, Jia Y, Tan O, Bourdette D, et al. Optical coherence tomography angiography enhances the detection of optic nerve damage in multiple sclerosis. Br J Ophthalmol. 2018; 102:520-524

-In light of these results, it may be hypothesized that a relationship exists between the decrease in the temporal and superior peripapillary RNFL and vasculature that nourishes this layer like Feucht et al [28] in his study that shows a high correlation between loss retinal vasculature and thinning retinal layers.

28-Feucht, N., Maier, M., Lepennetier, G., Pettenkofer, M., Wetzlmair, C., Daltrozzo, T, et al. Optical coherence tomography angiography indicates associations of the retinal vascular network and disease activity in multiple sclerosis. Multiple Sclerosis Journal. 2019;25(2), 224–234.

4. Line 102: “The control group consisted of subjects who did not have any type of relevant ocular or systemic disease.” The meaning of this sentence is not clear to the reviewer.

Response: Information added in methodology lines 109-112.

-The control group consisted of subjects who did not have any type of relevant ocular (epiretinal membranes, glaucoma, age related macular disease etc) or systemic disease previous were related with retinal vascular density such as Diabetes mellitus, arterial hypertension etc.

5. Line 116-117: “neurological and ophthalmologial examinations were…” should be combined into line 110 to clarify the meaning of “previous”.

Response: It has been added in line 119-121 new information to clarify the meaning of previous.

 - The previous neurlogical ophthalmological examination was used to detect ocular impairments such as glaucoma, cataract or media opacity that could affect functional vision or retinal microvascularization.

6. It is not clear to this reviewer whether the analysis was based on 6 x 6 mm scan for the area of 6 mm circular area with DTDRS partition, or for the area of 3 mm circular area. If this was based on the 3 mm ETDRS partition, why did the scan protocol use 6 x 6 mm. The authors will need to include a figure to show the scan area and partition.

Response: Triton OCTA protocol scan an area of 6x6mm but only cuantify percent of vessel density in area of ETDRS gird. It has been added an image (Line 157) of OCTA where it can show cross sectional Scan structural limited in red lines superficial vascular plexus and area analysed with the gird localization. 

Fig 1. Image of angiography optical coherence tomography measured in 6x6mm area in superficial vascular plexus with gird centered in macula. Gird is divided in five areas; central, nasal, interior, temporal and superior with number of percentages of vessel density.

7. Since both eyes in some cases were included, GEE will be needed to analyze the difference of the measurements between groups. If both eyes of the patient were treated independently, the statistic power was inflated, which is not right.

Response: In lines 165 to 169 we explain that following the Advised Protocol for OCT Study Terminology and Elements (APOSTEL) recommendations for reporting quantitative OCT studies, our study was performed using only one eye in MS patients (randomly selected), except in those patients in which only one eye had a history of optic neuritis (in these cases, both eyes were included in the analysis and were treated as independent) [20].

8. The authors need to report the cut off value of scan quality scores.

Response: In line 141-144. All measurements were taken by a single observer and low-quality ,and only images with signal strength index (SSI) and analyzed images with quality score above 50 and 40 respectively [19] were included and also images with movement artefacts were excluded from the analysis.

19. Tang, FY, Erica O.Chan, Sun Z, Wong R, Lok J, Szeto S et al. Clinically relevant factors associated with quantitative optical coherence tomography angiography metrics in deep capillary plexus in patients with diabetes. Eye and vision. 2020;7:7.

9. The superficial vascular plexuses (SVP) appeared to be defined as the vasculature from ILM to RPE, which is the vascular network of the entire retina, not the only SVP. The authors will need to separate SVP and deep vascular plexus for analysis and comparison. In addition, the authors will need to include OCTA enface view image and cross-sectional structural image with the segmented boundaries.

Response; We changed the limitation of SVP in line 149. Also, it has been added an image of OCTA which shows a cross sectional Scan structural limited by red lines of the superficial vascular plexus and area analysed with the gird localization. 

.

-Vessel density values were calculated from the internal limiting membrane to the retinal pigment epithelium inner plexiform layer.

-Fig 1. Image of angiography optical coherence tomography measured in 6x6mm area in superficial vascular plexus with gird centered in macula. Left image shows a B-scan with orange lines limiting analyzed zone. Right image shows density map with gird divided in five areas; central, nasal, interior, temporal and superior with number of percentages of vessel density.

10. Since the OCTA can get the thickness information of intraretinal layers, the authors need to report the thickness information to show whether structural thinning occurred as the author discussed this relation in Discussion.

Response; OCTA device does not provide structural measures of the area measured with this protocol. But we propose this in limitations (line 346).

- We expose a retinal vessel density loss in FM patients without consider structural retinal state, this device does not measure thickness in same mensuration of OCTA, but more information about thickness of the retinal layers would complete the study.

11. Again, if individual thinning of the retina occurred, the loss of vessels may be proportional to the loss of tissue. Therefore, normalization of the vessel loss needs to be done by the vessel density divided by the corresponding tissue volume. This will provide more accurate information whether hypoperfusion occurs. The absolute value decrease of the vessel percentage without knowing the perfused tissue mess in volume does not mean hypoperfusion.

Response; In order of your observation, we have written less of VD instead of hypoperfusion vascular. Also, we added this limitation in our study. Triton OCTA device does not provide structural measures of the area measured with this protocol. But in discussion we propose that it would be interest to analyse structure of superficial layers with same OCT device to know how the correlation between vasculature and structural area is.

Line 348: We expose a retinal vessel density loss in FM patients without consider structural retinal state, this device does not measure thickness in same measurement as OCTA, but more information about thickness of the retinal layers would complete the study. 

12. The authors will need to discuss the mechanism of localized alterations of the VD in the whole group and subgroups. Previous studies showed more severe loss of VD in MS+ON. It is surprised that there appeared to be no difference of the VD measurements between MS+ON and MS-ON. The authors explained this to be due to small sample size. Increasing the sample with MS+ON will be needed.

Response; We related the results about not differences between MS+ON and MS-ON due only eyes with neuritis episode before more than one year were included and the results show that there are not vascular sequelae after one year of neuritis episode. In the other hand, sample was small so we added that in limitations of the study the small sample size.

- Line 339: This presents a challenge in patients with debilitating diseases, especially in those suffering severe cognitive and physical impairments that make it difficult for them to follow instructions and maintain concentration and this affected the small sample size presented in this study.

Reviewer #3: 

Beatriz Cordón Ciordia and colleagues provide a paper on OCTA vessel density in patients with MS and compared their data with normals.

Introduction and method section is finely written. The authors should include here exemplary vessel density en face pictures with an overlay of the grit used to better understand the position of the values. Especially central macula should be highlighted as normally this area is the foveal avascular zone without any vessels.

Response; It has been added an image of OCTA where it can show cross sectional Scan structural limited in red lines superficial vascular plexus and area analysed with the gird localization. 

Fig 1. Image of angiography optical coherence tomography measured in 6x6mm area in superficial vascular plexus with gird centered in macula. Left image shows a B-scan with orange lines limiting analyzed zone. Right image shows density map with gird divided in five areas; central, nasal, interior, temporal and superior with number of percentages of vessel density.

The results are well displayed and correlated. The graph should show the results in boxplots to better visualize the distribution of the results in normals and MS patients.

-Figure 2 has been modified and changed to boxplots to better visualize the distribution of the control and MS patients. 

Fig 2. Graphic of percentage of superficial plexus macular in both groups. A lower vascular density tendency is observed in the group of MS patients, with significance existing in the nasal, inferior and superior macular areas.

The discussion should include an explanation why nasal and superior sectors might be correlating so well and disctinct from the normal population. 

Response: Line 297; García-Martín et al [27], conduct five-year follow-up of MS patients, observed that the retinal nerve fiber layer and the ganglion cell layer measured in the peripapillary ring are thinner in MS patients, especially in the temporal and superior areas.In light of these results, it may be hypothesized that a relationship exists between the decrease in the temporal and superior peripapillary RNFL and vasculature that nourishes this layer like Feucht et al [28] in his study that shows a high correlation between loss retinal vasculature and thinning retinal layers.

27- Adhya S, Johnson G, Herbert J, Jaggi H, Babb JS, Grossman RI, et al. Pattern of hemodynamic impairment in multiple sclerosis: dynamic susceptibility contrast perfusion MR imaging at 3.0. T. Neuroimagen. 2006; 33 (4): 1029-1035.

28-Feucht, N., Maier, M., Lepennetier, G., Pettenkofer, M., Wetzlmair, C., Daltrozzo, T, et al. Optical coherence tomography angiography indicates associations of the retinal vascular network and disease activity in multiple sclerosis. Multiple Sclerosis Journal. 2019;25(2), 224–234.

In MS RNFL in the papillomacular bundle shows loss of thickness in structural OCT already in very early stages. These axons belong mainly to midget or P-ganglion cells. Does the OCTA device provide structural measures of the areas measured with OCTA? if yes, the authors should include these measurements and correlate them too.

Response; OCTA device does not provide structural measures of the area measured with this protocol. But we propose this in limitations (line 346).

- We expose a retinal vessel density loss in FM patients without consider structural retinal state, this device does not measure thickness in same mensuration of OCTA, but more information about thickness of the retinal layers would complete the study.

The authors should correlate vessel density measures with their data as most studies were performed with Optovue and this study was done with a Topcon device. As all OCTA devices use different algorithms and the data are hardly comparable the authors should also take this methodological aspect into account.

Response; Results have been related with other similar study realized with Optovue device in line 294

- In this study, a significant decrease in vascular flow VD was marked in the superior, nasal and inferior macular areas were observed in patients suffering from MS. Lanzillo R et al [12] show a loss of VD in all sectors of macular ETDRS measured with Octovue device in MS patients with higher EDSS values than sample of this study.

submitted the response to reviewers and a new manuscript with changes and corrections. 

Thank you very much for improving our paper.

Sincerely, 

 B. Cordón Ciordia

---

## [Decision Letter · Decision Letter 1]

30 Jul 2020

PONE-D-20-14099R1

Angiography with optical coherence tomography as a biomarker in multiple sclerosis

PLOS ONE

Dear Dr. Ciordia,

Thank you for submitting your manuscript to PLOS ONE. After careful consideration, we feel that it has merit but does not fully meet PLOS ONE’s publication criteria as it currently stands. Therefore, we invite you to submit a revised version of the manuscript that addresses the points raised during the review process.

We look forward to receiving your revised manuscript.

Kind regards,

Ireneusz Grulkowski, PhD

Academic Editor

PLOS ONE

Journal Requirements:

Additional Editor Comments (if provided):

Please, address reviewer's concerns.

Reviewers' comments:

Reviewer's Responses to Questions

**Comments to the Author**

1. If the authors have adequately addressed your comments raised in a previous round of review and you feel that this manuscript is now acceptable for publication, you may indicate that here to bypass the “Comments to the Author” section, enter your conflict of interest statement in the “Confidential to Editor” section, and submit your "Accept" recommendation.

Reviewer #1: All comments have been addressed

Reviewer #2: (No Response)

2. Is the manuscript technically sound, and do the data support the conclusions?

Reviewer #1: Yes

Reviewer #2: No

3. Has the statistical analysis been performed appropriately and rigorously? 

Reviewer #1: Yes

Reviewer #2: N/A

4. Have the authors made all data underlying the findings in their manuscript fully available?

Reviewer #1: Yes

Reviewer #2: Yes

5. Is the manuscript presented in an intelligible fashion and written in standard English?

Reviewer #1: Yes

Reviewer #2: No

6. Review Comments to the Author

Reviewer #1: (No Response)

Reviewer #2: Thank you the effort the author team made to address these comments and providing the datasheet. I studied the revision and datasheet with great interest in understanding the study and conclusion. Please see the details in the attachment.

7. PLOS authors have the option to publish the peer review history of their article (what does this mean?). If published, this will include your full peer review and any attached files.

Reviewer #1: No

Reviewer #2: No

---

## [Author Response · Author response to Decision Letter 1]

8 Sep 2020

- Thank you for your effort to thoroughly review our study. We are grateful for your great involvement with this work. All measurements were taken by a single observer, and only images with signal strength index (SSI) and analyzed images with quality score above 50 and 40 respectively were included and also images with movement artefacts were excluded from the analysis.

There is some controversy in the removal or inclusion of outlier data. In our case, delete the most extreme data modifies the content of the article and we want to remain faithful to our study, because this data meets the inclusion and exclusion criteria of quality scores of our study in addition to the fact that it has already been accepted by the two previous reviewers. It is not intended to hide information since these data can be seen reflected in the box diagram of figure 2. However, we are willing, if you consider it, to add a following column to our results with the significance that results from eliminating the data outliers. Eliminating these data, the macular upper quadrant continues to have significant differences between MS patients and healthy controls.

- We have added a clarifying phrase to add the number of patients who did the quality of life questionnaire MSQoL-54 and also, we add this in limitations of the study.

o Line 203: MSQoL-54 was completed by 20 MS patients.. 

o Line 348: The number of patients who answered MSQoL-54 was small compared to the total sample.

- We have added these recent papers in our discussion

o Line 299: In this study, a significant decrease in VD was observed in the superior, nasal and inferior macular areas in patients suffering from MS with Triton Plus OCTA. Our results support previous research, such as Lanzillo R et al [12] (with Octovue OCTA device) and Olwen C Murphy et al [29] (Spectralis OCTA), both showing a loss of VD in SVP macular ETDRS in MS patients. Only one study evidenced an increase in VD in MS patients [30], which was realized with Angioplex Zeiss device.

Thank you for your reviewers to improve quality of this study.

We hope that these changes work out your doubts.

---

## [Decision Letter · Decision Letter 2]

17 Sep 2020

PONE-D-20-14099R2

Angiography with optical coherence tomography as a biomarker in multiple sclerosis

PLOS ONE

Dear Dr. Ciordia,

Thank you for submitting your manuscript to PLOS ONE. After careful consideration, we feel that it has merit but does not fully meet PLOS ONE’s publication criteria as it currently stands. Therefore, we invite you to submit a revised version of the manuscript that addresses the points raised during the review process.

We look forward to receiving your revised manuscript.

Kind regards,

Ireneusz Grulkowski, PhD

Academic Editor

PLOS ONE

Additional Editor Comments (if provided):

Please, revise the manuscript and address the issues by extending the discussion and modifying the table.

Reviewers' comments:

Reviewer's Responses to Questions

**Comments to the Author**

1. If the authors have adequately addressed your comments raised in a previous round of review and you feel that this manuscript is now acceptable for publication, you may indicate that here to bypass the “Comments to the Author” section, enter your conflict of interest statement in the “Confidential to Editor” section, and submit your "Accept" recommendation.

Reviewer #1: (No Response)

Reviewer #2: (No Response)

2. Is the manuscript technically sound, and do the data support the conclusions?

Reviewer #1: (No Response)

Reviewer #2: Partly

3. Has the statistical analysis been performed appropriately and rigorously? 

Reviewer #1: (No Response)

Reviewer #2: No

4. Have the authors made all data underlying the findings in their manuscript fully available?

Reviewer #1: (No Response)

Reviewer #2: Yes

5. Is the manuscript presented in an intelligible fashion and written in standard English?

Reviewer #1: (No Response)

Reviewer #2: No

6. Review Comments to the Author

Reviewer #1: (No Response)

Reviewer #2: Thank you for the revision. I have some recommendations:

1. As the authors indicated in their responses to the issue of outliers, this reviewer believes that it is necessary to include new columns (mean ± SD and P value) of the control group and P values in comparison of the MS group in Tables 3, 4, 5 and 6.

2. The authors indicated that the macular upper quadrant continues to have significant differences between MS and controls after eliminating the data outliers. However, the t-test with two tails yielded P value as 0.495, a marginal significant level. Recommend to further discuss.

3. In addition, to better interpret the data, averaged vascular density of the macular area (averaged from 4 quadrants) needs to be included in these tables, which will conclude whether the vascular density altered in the macular area in patients with MS. This will also facilitate the comparison of their work to the previously published work.

4. Recommend to include a scatterplot (subject sequential ID vs measurements) in addition to Figure 2. Figure 2 does not show the sequence of data collection. The authors will also need to discuss the possible reasons why the outliers happened in the data’s early collection

7. PLOS authors have the option to publish the peer review history of their article (what does this mean?). If published, this will include your full peer review and any attached files.

Reviewer #1: No

Reviewer #2: No

---

## [Author Response · Author response to Decision Letter 2]

21 Oct 2020

Thank you again for your revision. We review and explain your suggestions one by one below. 

We would like the reviewers to clarify where the paragraphs considered incorrect in English redaction are. This paper was reviewed by an English native editor but we can edit the manuscript again for a better understanding. 

1. As the authors indicated in their responses to the issue of outliers, this reviewer believes that it is necessary to include new columns (mean ± SD and P value) of the control group and P values in comparison of the MS group in Tables 3, 4, 5 and 6.

As the reviewer proposed here, we have included new data in tables 3, 4, 5 and 6: mean ± SD value in control and EM groups without outliers and P value with these data.

2. The authors indicated that the macular upper quadrant continues to have significant differences between MS and controls after eliminating the data outliers. However, the t-test with two tails yielded P value as 0.495, a marginal significant level. Recommend to further discuss.

In the superior macular quadrant, p value is 0.047 after removing outliers.

Line 219: Superior macular area remains minimally significantly low vascular density in EM groups despite removing the outlier (p =0.047). 

Line 312: More precisely, VD decrease remains significant when we remove outliers.

Line 320: In light of these results, it may be hypothesized that a relationship exists between the decrease in the temporal and superior peripapillary RNFL and vasculature that nourishes this layer, especially in superior macular area that shows a marked decrease in our results. Feucht et al [33] in his study shows this with a high correlation between loss retinal vasculature and thinning retinal layers.

3. In addition, to better interpret the data, averaged vascular density of the macular area (averaged from 4 quadrants) needs to be included in these tables, which will conclude whether the vascular density altered in the macular area in patients with MS. This will also facilitate the comparison of their work to the previously published work.

A row was added with average vascular density of the 4 quadrants of the macular area in tables 3, 4, 5 and 6 for a better interpretation the data.

4. Recommend to include a scatterplot (subject sequential ID vs measurements) in addition to Figure 2. Figure 2 does not show the sequence of data collection. The authors will also need to discuss the possible reasons why the outliers happened in the data’s early collection

A scatterplot of average macular vascular density was added as Figure 2. Studies like Orlov et al, show that there is a variability in retinal microvasculature due to age. In our study variability in values of vascular density is present but we consider this variability representative of a normal population sample and only images of high quality were used in our study.

Line 210: It is necessary to consider that retinal vasculature has some anatomical variations sensitive to axial length, refraction, age… (26), which causes variability in vascular density values. Figure 2 show VD values in control group and EM group related with age. (Fig 2)

Fig 2. Scatterplot graphic of vascular density and age in control group and MS group.

26- Orlov NV, Coletta C, van Asten F, Qian Y, Ding J, AlGhatrif M, et al. Age-related changes of the retinal microvasculature. PLoS One. 2019 May 2;14(5): e0215916.

---

## [Decision Letter · Decision Letter 3]

18 Nov 2020

Angiography with optical coherence tomography as a biomarker in multiple sclerosis

PONE-D-20-14099R3

Dear Dr. Ciordia,

We’re pleased to inform you that your manuscript has been judged scientifically suitable for publication and will be formally accepted for publication once it meets all outstanding technical requirements.

Kind regards,

Ireneusz Grulkowski, PhD

Academic Editor

PLOS ONE

Additional Editor Comments (optional):

Reviewers' comments:

Reviewer's Responses to Questions

**Comments to the Author**

1. If the authors have adequately addressed your comments raised in a previous round of review and you feel that this manuscript is now acceptable for publication, you may indicate that here to bypass the “Comments to the Author” section, enter your conflict of interest statement in the “Confidential to Editor” section, and submit your "Accept" recommendation.

Reviewer #2: All comments have been addressed

2. Is the manuscript technically sound, and do the data support the conclusions?

Reviewer #2: Yes

3. Has the statistical analysis been performed appropriately and rigorously? 

Reviewer #2: Yes

4. Have the authors made all data underlying the findings in their manuscript fully available?

Reviewer #2: Yes

5. Is the manuscript presented in an intelligible fashion and written in standard English?

Reviewer #2: Yes

6. Review Comments to the Author

Reviewer #2: Thank you very much for answering all the questions. All my comments have been addressed. No further suggestions.

7. PLOS authors have the option to publish the peer review history of their article (what does this mean?). If published, this will include your full peer review and any attached files.

Reviewer #2: No

---

## [Editor Report · Acceptance letter]

25 Nov 2020

PONE-D-20-14099R3 

Angiography with optical coherence tomography as a biomarker in multiple sclerosis 

Dear Dr. Cordon:

I'm pleased to inform you that your manuscript has been deemed suitable for publication in PLOS ONE. Congratulations! Your manuscript is now with our production department. 

Kind regards, 

on behalf of

Dr. Ireneusz Grulkowski 

Academic Editor

PLOS ONE